# Applicable Plant Proteins and Dietary Fibers for Simulate Plant-Based Yogurts

**DOI:** 10.3390/foods10102305

**Published:** 2021-09-28

**Authors:** Jae-Sung Shin, Beom-Hee Kim, Moo-Yeol Baik

**Affiliations:** 1Department of Food Science and Biotechnology, Institute of Life Science and Resources, Kyung Hee University, Yongin 17104, Korea; drumlover@naver.com (J.-S.S.); qjagml5567@naver.com (B.-H.K.); 2Corporate Technology Office, Pulmuone Corp., Cheongju 28614, Korea

**Keywords:** stirred soy yoghurt, plant protein, dietary fiber, clean label, physical properties

## Abstract

Effects of plant proteins and dietary fibers on the physical properties of stirred soy yogurt were investigated. Buffering capacity against lactic acid was not affected by the protein concentration for any of the four proteins that were examined: isolate soy protein (ISP), pea protein (PP), rice protein (RP), and almond protein (AP). Three proteins other than AP exhibited an increase in buffering capacity (dB/dPH) following a physical treatment, whereas AP saw a decrease in buffering capacity. Furthermore, physically treated PP revealed a significant increase in viscosity, reaching up to 497 cp in the pH 6.0~6.2 range during the titration process. Following fermentation, PP produced the highest viscosity and coagulum strength with no syneresis. In the case of dietary fiber, Acacia Fiber (AF) was completely dissolved in the solvent and did not affect the physical properties of the fermented coagulum. Soy fiber (SF) was also not suitable for fermented milk processes because precipitation occurred after the physical treatment. In the case of citrus fiber (CF), however, syneresis did not occur during storage after the physical treatment, and the viscosity also increased up to 2873 cP. Consequently, PP and CF were deemed to be a suitable plant protein and dietary fiber for stirred soy yogurt, respectively.

## 1. Introduction

Korea’s annual market size for fermented milk products ranges from approximately $764 million to $850 million. Among such products, the stirred yogurt market has experienced continuous growth in recent years and now accounts for 43% of all fermented milk product sales. This has resulted in a steady shift away from the previously dominant drink yogurt market. However, the overall fermented milk market has showed slow growth over the past few years, resulting in an over-competitive market where players compete through prices rather than product quality. As a result, there is a need to help consumers look towards new market categories rather than existing, over-competitive markets [1].

Young Korean millennials have great interest in consumption value, vegetarianism, and green dining. Approximately 1.5 million people in Korea prefer a vegetarian diet, and the vegan population is estimated to be 500,000: approximately 3% of the Korean population and a tenfold increase in the past ten years [2]. Since the nation implemented a certification system for vegan products in the first quarter of 2018, a total of 364 products have been certified and registered by the Korea Agency of Vegan Certification and Services in of the first quarter of 2021. This highlights the rapid growth of the vegan market. The global plant-based food market size was valued at $42,186.43 million in 2020 and its 66% was the alternatives for dairy foods. Recently, many consumers have shifted towards veganism, and the benefits associated with plant-based foods are the primary driving force behind the growth of the industry [3].

The demand for yogurts made with vegetable ingredients is rising as an alternative to milk-based yogurt, which is made with animal milk. In particular, significant research has focused on soy yogurt, which is produced by fermenting soy milk [4,5,6,7,8,9]. However, most studies involve animal experiments to investigate the effects of soy yogurt; only a few studies have reported results regarding the physicochemical changes that arise from physical treatments. Physicochemical, textural, and volatile flavor compound changes according to fermentation has been reported [4]. It has been reported that raffinose, which is the source of the unpleasant odors of soy beans, decomposes during the fermentation of soy milk [5,6]. Response surface methodology (RSM) analysis was applied to optimize fermentation conditions according to prebiotic and probiotic content [7]. Effect of ultra-high-pressure homogenization conditions on the storage quality of soy yogurt was reported [8], and the effect of ultra-high-pressure treatment on the quality-related physical properties of soy yogurt was investigated [9]. It is clear that most research efforts have pointed towards a study of the properties of soy yogurt according to the fermentation or treatment conditions. In contrast, there are very few studies on the properties of stirred soy yogurt.

The most important characteristics of stirred yogurt is that it must have qualities that make it suitable for consumption using a spoon. Simply fermenting soy milk results in a yogurt with a low viscosity, which is undesirable for stirred yogurt. Additives such as thickeners (hydrocolloids, emulsifiers, starch) are the simplest means of further enhancing the properties that make yogurts suitable for spoon consumption [7,9]. However, the use of additives conflicts with the ideals of vegan and green dining, which leans towards the clean label movement. Fortunately, there are alternative methods of producing yogurts with more suitable properties. For example, coagulum strength and viscosity can be increased by using the properties of proteins that are coagulated by lactic acid, which is produced when lactobacillus undergoes fermentation. Additionally, the water binding in a yogurt can be efficiently controlled by adding clean label dietary fibers, which further enhances the preferable properties. As such, this study evaluates the physicochemical properties of plant proteins and dietary fibers that can be used to produce clean label stirred soy yogurt free of chemical additives.

## 2. Materials and Methods

### 2.1. Materials

This study tested four different plant proteins. DORECKON SOY 200 (Dezhou Ruikang Bio Technology Co., LTD, Dezhou City, China) with at least 90% protein content was used to source of isolated soy protein (ISP) and Empro E86 (EMSLAND-Strarke GmbH, Emlichheim, Germany) with at least 84% protein content was used to source of pea protein (PP). For rice protein (RP), RP-80 (VEDAN Enterprise Corp., LTD, Dong Nai, Vietnam) with at least 80% protein content was used, and almond protein powder (Blue Diamond Growers, Sacramento, CA, USA) with 40–45% protein content was used to source of almond protein (AP).

Three different plant-based dietary fibers were used. Soy fiber (SF) was sourced from YX100(s) (SHANDONG YUXIN Bio-Tech Co., LTD, Qingdao, China) with 13~16% protein content and 13~26% dietary fiber content. For citrus fiber (CF), we used Citri-Fi 100 (Fiberstar Co., LTD, River Falls, WI, USA). Acacia fiber (AF) was sourced from Fibregum B (Colloides natuarels international Co., LTD, Rouen, France) with at least 90% dietary fiber content.

For the lactic acid bacteria (LAB) used in the fermentation experiments, the VEGE-033 Culture (DuPont^TM^ Danisco^R^, København, Denmark), which is suitable for vegetarians, was used. For dextrose monohydrate, which is used to grow LAB, Daesang Dextrose Monohydrate (DAESANG Co., Incheon, Korea) were used.

Lactic acid (KANTO Chemical Co., Inc., Tokyo, Japan) and 0.1 N sodium hydroxide standard solution (CAS N0. 1310-73-2, DAEJUNG Chemicals & Metals Co., LTD, Siheung, Korea) were used. Distilled water from a water purification system (Barnstead diamond T2, Thermo fisher scientific, Waltham, MA, USA) was used.

### 2.2. Sample Preparation

#### 2.2.1. Protein Dispersions for Titration Capability

During fermented milk manufacturing processes, lactic acid content increases due to the fermentation of LAB. The change in the titration volume of each protein to neutralize the lactic acid was analyzed with 1 N NaOH for increasing amounts of protein powder. The suspensions for the experiment were prepared by adding the protein powders to distilled water that was stored at 25 °C to produce suspensions of various concentrations ranging from 1–10% (*w*/*w*). A series of suspensions were prepared for each protein powder, and the suspensions were thoroughly mixed using a hand blender (MQ 120, Hi-P Poland Sp. Z.o.o., Magazynowa, Poland).

#### 2.2.2. Protein Dispersions for Buffering Capacity

Fermented milk manufacturing processes require the physical treatment stages of pasteurization and homogenization. The following protein suspensions were prepared to investigate the role of the physical treatment on the buffer capabilities of the proteins. Each protein powder was poured in 25 °C distilled water and thoroughly mixed with a hand blender to make 5% (*w*/*w*) suspensions. A portion of the prepared suspensions was placed in a cooler and was used to measure the properties without the physical treatment (pasteurization and homogenization). The remaining suspensions were warmed to 65 °C and were homogenized using a laboratory homogenizer (APV-1000, APV Gaulin, Assen, The Netherlands) under a first stage pressure of 140 bar and a second stage pressure of 20 bar (total pressure: 160 bar). The homogenized suspensions were placed in glass containers and immediately sterilized by heating to 95 °C for five minutes. After the pasteurization process, the suspensions were placed in a cooler. The properties of the treated and untreated samples were measured and compared to analyze the effects of the physical treatment.

#### 2.2.3. Fermentation of Vegetable Proteins

To investigate the role of lactic acid produced by LAB on formation and properties of coagulum, samples were prepared as follows. Protein powder (5%, *w*/*w*) and dextrose monohydrate (2% *w*/*w*) were thoroughly mixed in 65 °C distilled water using a hand blender. The mixed samples were homogenized and sterilized as previously described and were subsequently cooled to 40 °C for LAB inoculation. The LAB strain VEGE-033 was diluted 100 times in a sterile solution of 0.85% NaCl and was subsequently mixed with the sterilized and cooled samples with a ratio of 2 mL/kg (*v*/*w*). The LAB and sample mixtures were gently shaken to ensure sufficient mixing.

The mixtures were divided into two experimental groups. The first group was used to measure the coagulum strength. Here, the mixtures were poured in 150 g cups and were then sealed using a heat-press sealer. The samples were then incubated in an incubator for eight hours and stored in a cooler at 5 °C for 24 h, after which the coagulum strength was measured. The samples of the other group, which were used to measure the fermentation characteristics, were transferred to sterilized 1 L glass bottles and placed in a water bath (BS-21, Jeio Tech, Seoul, Korea) for incubation at a constant temperature of 40 °C: the fermentation temperature of LAB. Following incubation for eight hours, the fermented broth was gently shaken to break down the coagulum and prevent the LAB from further fermentation. The mixtures were then cooled until the coagulum temperature reached 20 °C. The fermented broth mixtures were sieved through a 40 mesh filter to further break down the coarse coagulum lumps into a smooth residue, which was divided into 150 g cups and sealed using a heat-press sealer. The cups were stored in a cooler at 5 °C to carry out the further experiments.

#### 2.2.4. Dietary Fiber Dispersions

The dietary fiber suspensions were prepared as follows. Each dietary fiber was thoroughly mixed in 65 °C distilled water with a blender to produce mixtures with a concentration of 5% (*w*/*w*). The mixtures were homogenized and sterilized as previously described and then transferred to 50 mL conical tubes. The conical tubes were stored in a cooler at 5 °C to carry out the next experiments.

### 2.3. Buffering Capacity of Vegetable Proteins

#### 2.3.1. Titration Capability

Titration was performed to measure the amount of 1 N lactic acid (mL) required to reduce the pH of 1~10% (*w*/*w*) protein suspensions (200 g) to 4.2, which is the general end point of fermentation for fermented milk. Then, the effect of the protein concentration on the lactic acid titration volume within the pH range at which fermentation takes place was investigated.

#### 2.3.2. Buffering Capacity

An experiment was conducted to measure the change in the buffering capacity of the proteins upon physical treatment. The initial pH levels of the samples that were homogenized, sterilized, and cooled, as well as the untreated samples, were measured. Then, we performed titration with 1 N lactic acid and compared the change in the lactic acid titration volume. To better compare the properties of each protein, the end point of titration pH was set as pH 3.0, which is lower than the general pH level of fermented milk manufacturing processes. The protein powder concentration was set as 5% (*w*/*w*) for all samples to exclude protein concentration as a variable.

The buffering capacity was calculated for each pH region in the titration curve to analyze the differences between the treated and untreated samples for each region. The buffering capacity for each pH region can be measured by finding the slope of the tangent of the curve for each region. The dB/dpH ratio, which is used to calculate the buffering capacity for a pH region, represents the relationship between the volume of acid or base that is added and the change in pH [10].
(1)$$\frac{\mathrm{dB}}{\mathrm{dpH}}=\frac{\left(\mathrm{volume}\;\mathrm{of}\;\mathrm{acid}\;\mathrm{or}\;\mathrm{base}\;\mathrm{added}\right)\times \left(\mathrm{normality}\;\mathrm{of}\;\mathrm{acid}\;\mathrm{or}\;\mathrm{base}\right)}{\left(\mathrm{volume}\;\mathrm{of}\;\mathrm{sample}\right)\times \left(\mathrm{pH}\;\mathrm{change}\;\mathrm{produced}\right)}$$

### 2.4. Physical Properties

#### 2.4.1. pH

The pH values of the samples were measured using a pH meter (Thermo Scientific Orion 3 Star, Thermo Electron Corporation, Beverly, MA, USA) with ROSS Ultra pH/ATC Triode (8157 BNUMD) electrodes from the same manufacturer.

#### 2.4.2. Titratable Acidity (TA)

The lactic acid produced by fermentation was titrated using a 0.1 N sodium hydroxide standard solution (factor = 1). Titratable acidity (TA) was calculated using the following equation:TA (%) = a × f × 0.009/“g” of used sample(2)
where, a = volume (mL) of 0.1 N NaOH used for titration, f = factor of 0.1 N NaOH.

#### 2.4.3. Coagulum Strength

Coagulum strength was measured using a Sun Rheometer (CR-3000EX-S, Sun Scientific CO., LTD, Tokyo, Japan). A No. 1 30 mm-diameter probe made specifically for pressure and elasticity measurements was used, and the velocity of the table was set at 60 mm/min. The distance of approach from the surface of the coagulum was set at 15 mm.

#### 2.4.4. Apparent Viscosity

The apparent viscosities of the samples were measured using a Brookfield Viscometer (DV2T, AMETEK Brookfield, Middleboro, MA, USA). Spindles for liquid viscosity measurement were used in the experiment; a No. 62 spindle was used below 1000 cp and a No. 63 spindle was used above 1000 cp. Viscosity changes were measured as the spindles were rotated at 30 rpm for one minute.

#### 2.4.5. Syneresis

Syneresis of proteins due to the formation of coagulums from fermentation was determined. Samples with or without inoculation of LAB were transferred to 50 mL conical tubes and left to ferment at 40 °C for 12 h. Afterwards, the samples were stored in a cooler at 5 °C, and syneresis was measured on the third day of storage.

To analyze the syneresis according to the dispersibility of the dietary fibers, the dietary fiber suspensions were poured in 50 mL conical tubes and stored in a cooler at 5 °C, then syneresis was measured on the third day of storage. In both cases, syneresis was determined using a mass cylinder. After storage, supernatants were carefully transferred to a mass cylinder using a pipette and determined the volume of transferred supernatant.

### 2.5. Color and Soluble Solid Content

Chromaticity was measured using a spectrophotometer (CM-3500d, Konica Minolta Sensing, INC, Osaka, Japan). Differences in color were recorded in CIE L*a*b* scale in terms of lightness (L*) and color (a*—redness; b*—yellowness). The total color difference (ΔE) and whiteness index (WI) was calculated.
(3)$$\Delta E=\sqrt{{\Delta L}^{2}+{\Delta a}^{2}+{\Delta b}^{2}}$$
(4)$$\mathrm{WI}=100-\sqrt{{\left(100-L\right)}^{2}+a^{2}+b^{2}}$$

Additionally, soluble solid content was measured using a digital refractometer (RX-500 α, ATAGO CO., LTD, Tokyo, Japan).

### 2.6. LAB Dertermination

LAB was determined using the following method. Samples (10 g) were added into 90 mL of sterilized saline solution and vortexed for 1 min. Samples were serial diluted up to 10^8^. The diluted sample (1 mL) was mixed with sterilized MRS medium (20 g) and poured into a disposable petri dish. Samples are incubated in 37 °C incubator (BF-18i, BNF Korea, Seoul, Korea) for 48 h and the colony number is counted. Consequently, LAB was calculated using a dilution factor.

### 2.7. Statistical Analysis

All experiments were performed in triplicates. All test data were analyzed using analysis of variance (ANOVA) and were expressed as mean values ± standard deviations. Tukey’s test was applied to determine the significance of differences in variables among different treatment groups at a significance level of 95% (α < 0.05) using SAS version 8.02 for Windows (SAS Institute, Inc., Cary, NC, USA).

## 3. Results and Discussion

### 3.1. Buffering Capacity of the Vegetable Proteins

#### 3.1.1. Titration Capability

Figure 1 shows the amount of 1 N lactic acid that was titrated with increasing concentrations of untreated protein suspensions to reach pH 4.2, which is the general end point for the fermentation of fermented milk. For all proteins used in the experiment, the amount of lactic acid increased proportionally to the protein concentration, indicating that protein concentration did not affected the change in the amount of lactic acid consumed. The lactic acid titration volume increased proportionally to the protein concentration for ISP, PP, and AP with relatively higher correlation coefficient (R^2^ > 0.98). In the case of RP, which had an initial pH of 4.8, the pH level decreased to below 4.2 only after 0.1~0.3 mL of 1 N lactic acid was added. As such, it was not possible to obtain valid titration results according to the protein concentration. ISP and PP had slopes of 1.7264 and 1.8404, respectively, which are similar values to the slopes of proteins from the legume family. AP, which is a type of nut, had a slope of 1.4151 and required less lactic acid for titration than ISP or PP.

In general, dairy products include various components that act as buffers against acidity. These components consist of small compounds that contain proteins with one or more acid-base group. The buffering capacity of dairy products is equal to the combined buffering capacities of individual acid-base groups, but since each component is not “free” in the solution, interactions between the compounds must be taken into account [11]. This is also true for plant proteins: an increase in the protein powder concentration without any changes to the constituents leads to an increase in the total buffering capacity ratio. As such, a strong correlation is likely between the protein powder concentration and the lactic acid titration volume.

The buffering capacities of dairy products have been reported to vary depending on the species of livestock from which the product is obtained [12,13]. Such differences are due to the qualitative and quantitative compositional variations of milk obtained from different species of livestock. More specifically, the buffering capacities of milk obtained from different species of livestock species are dependent on the protein content, colloidal calcium phosphate content, and highly phosphorylated casein content. Similarly, plant proteins also vary in terms of acid-base composition and structure depending on the plant species, which may explain the variations in the slopes of the protein powder concentration–lactic acid titration volume curves.

#### 3.1.2. Buffering Capability

Table 1 shows the initial pH and the titration amounts of lactic acid for each protein before and after pasteurization and homogenization. According to the change in initial pH following pasteurization and homogenization, the initial pH of ISP and PP did not change after the homogenization/pasteurization treatment, whereas RP and AP showed increases in the initial pH following the treatment. This indicates that the RP and AP proteins underwent greater structural changes due to the physical treatment compared to ISP and PP. The pH of almond milk derived from an almond and water mixture with a 1:4 ratio was 6.2 without the physical treatment [14], and the pH of homogenized/sterilized almond milk was 6.64 [15]. Similar to the aforementioned research results, it is believed that the increase in pH of AP is due to the significant structural changes of the protein caused by the pasteurization and homogenization processes.

For all proteins, the lactic acid titration volumes of proteins that had undergone the physical treatment (homogenization and pasteurization) were greater than the untreated proteins. Specifically, the lactic acid titration volume increased by 9.2% for ISP, 17.3% for PP, 37.5% for RP, and 11.2% for AP. Although the largest increase in titration volume occurred for RP as the amount of 1 N lactic acid required for titration was miniscule compared to the other protein cases, statistical analysis showed that the resulting change in lactic acid titration amount was not significant. Conversely, the other three proteins exhibited significant differences before and after the physical treatment with PP showing the largest increase in lactic acid titration volume.

Figure 2 shows the dB/dpH values of the proteins that represent the differences in the buffering capacities of physically treated and untreated proteins. Whereas ISP, PP, and RP exhibited increased buffering capacity upon physical treatment, the results of AP showed a decrease in buffering capacity. The proteins of the legume family, ISP and PP, share highly similar dB/dpH patterns. Without the physical treatment, the dB/dpH value increases with significant differences as the pH decreases. On the other hand, dB/dpH is higher for the physically treated proteins at high pH levels during the initial stages of the titration process. Furthermore, in the pH region above pH 5, the dB/dpH values are relatively similar. RP also exhibited an increase in buffering capacity following the physical treatment like the aforementioned proteins, but the small dB/dpH value did not show significant differences. In addition, the low volume of lactic acid used for titration resulted in a low dB/dpH value, indicating that the buffering capacity of RP is lower than the other proteins.

In the case of AP, the change in buffering capacity following pasteurization treatment shows a different trend compared to the other proteins, which may indicate that the pasteurization process caused AP to undertake greater structural changes compared to the other proteins. This resulted in the initial pH increasing from 6.1 to 6.6. Additionally, the dB/dpH value of AP at pH 5.0 and above is lower than the dB/dpH values at pH 6.5 for both the treated and untreated cases. This indicates a decrease in the buffering capacity of the protein within this pH region. In the pH region of 4.5–5.0, AP does not exhibit a significant increase in buffering capacity following the physical treatment as in the case of the other proteins. For pH 4.5 and below, the buffering capacity of the treated group is lower than the untreated group. Devnani et al. (2020) [14] reported that the stability of almond milk decreases at 55 °C and higher as almond milk proteins undergo an increase in protein surface hydrophobicity and molecular size and a decrease in alpha helix structures. Such an increase in surface hydrophobicity has been reported to be due to the exposure of the internal hydrophobic regions of amandine, the storage protein of almonds, as the protein molecule unfolds [16]. Similarly, the physical treatment is believed to reduce bonds between the AP molecules and to expose hydrophobic residues at the surface of the protein, resulting in a change in the initial pH and buffering capacity.

#### 3.1.3. Apparent Viscosity Changes According to pH following the Physical Treatment

The apparent viscosity changes were examined between treated and untreated groups. In the case of the untreated samples, the viscosity remained constant even as the pH changed due to the acid that was added during the titration experiment (data not shown). Conversely, the proteins in the sterilized and homogenized group exhibited viscosity changes according to pH.

Figure 3 shows the viscosity changes according to pH for each protein after the physical treatment, which decreased due to the addition of 1 N lactic acid for titration. PP exhibited an especially notable change in viscosity, increasing up to 497 cp in the 6.2–6.0 pH region. According to Klost et al. (2020) [17], the solubility of PP is the lowest at pH 6.0, and Klost and Drusch (2019) [18] reported that the rate of coagulum formation begins to increase at pH 6.2–6.5 as the pH decreases due to the acid produced from LAB fermentation. A maximum rate is achieved at pH 6.0, and then gradually decreases until pH 5.5. Such viscosity changes are expected to be useful for fermented milk manufacturing processes. The viscosity of fermented broths can be altered even by the small amounts of lactic acid produced during the early fermentation stages of the manufacturing process. This in turn increases the dispersion stability by preventing the precipitation of solids with a low solubility that can be produced in industrial-scale fermented milk manufacturing processes, allowing for the production of fermented milk with uniform properties. ISP also showed a slight increase in viscosity up to 127 cp in the pH 4.5–5.5 region. On the other hand, RP and AP exhibited relatively small changes in viscosity during the titration process that were insignificant and were within the measurement error range.

### 3.2. Fermentation Characteristics of Plant Proteins

#### 3.2.1. pH, TA and LAB

The physicochemical properties of the protein suspensions were analyzed following eight hours of fermentation. ISP formed coagulums after decreasing in pH through fermentation from an initial pH of 6.40 to 5.29, and the titratable acidity (TA) and number of LABs were 0.27% and 9.1 × 10^7^, respectively. PP showed a greater decrease in pH compared to ISP, decreasing from an initial pH of 7.30 to 5.61. PP also produced a TA of 0.28%. The number of LABs following fermentation was higher for PP than ISP at 1.68 × 10^8^ (Table 2).

RP produced a pH of 4.95 after fermentation, which was lower than ISP or PP. However, as its initial pH was 5.00, the overall pH change due to the fermentation of LAB was miniscule. The TA (0.09%) of RP was almost equal to the TA value of non-fermented sample (0.07%). Despite the initial number of LAB being 2.0 × 10^2^ cfu/g, no LAB was observed in the 10^3^–10^4^ times diluted solution that was used to measure the LAB in the fermented products, indicating almost no LAB growth. Furthermore, no coagulum formation was observed. In the case of RP, the initial pre-fermentation pH was 5.0 and the dB/dpH value until pH 4.0 was 0.11, a significantly lower value than the other proteins. Consequently, fermentation did not progress properly in RP, resulting in a lack of lactic acid being produced (Figure 2).

LABs used in this study are part of the vegan-friendly Vege series. According to the experimental results provided by the manufacturers of this type of LAB, fermentation in a solution with a low pH, such as 5~6, results in the pH remaining almost constant for the first six to eight hours, after which it begins to rapidly decrease (data not shown). They considered that the optimal growth condition for the lag phase of LAB is neutral. If the fermentation substrate is weakly acidic, LAB growth is hindered during the lag phase and delays the transition to the log phase. In this case, fermentation should be carried out for at least 10 h, yet this is not feasible for commercial fermented milk production due to the decreased production efficiency. Moreover, as pasteurization processes in fermented milk production do not achieve pasteurization, residual microorganisms are likely to grow in six to eight hours under the LAB fermentation temperature of 40 °C before the LAB enters its log phase, resulting in quality and safety concerns with the final product.

In the case of AP, although the protein exhibited a similar TA (0.29%) to ISP and PP, its pH (4.29) was lower than the aforementioned proteins. In other words, the H^+^ ion concentration in the solvent was relatively higher than the other proteins. Despite the lower pH of AP than ISP or PP, coagulum formation did not take place. This appears to be a result of the buffering capacity of the protein; according to the dB/dpH change before and after the physical treatment, the dB/dpH value (thus, the buffering capacity) decreases for AP in the pH range of 4.5–6.0, which is in contrast to ISP and PP where the buffering capacity increases as the pH decreases during fermentation (Figure 2). This results in an increase in the H^+^ ion concentration in the solvent and a lower pH for AP compared to ISP and PP.

According to a report involving a fermented milk production experiment using various types of milk, different types of milk have different buffering capacities: goat milk possesses the lowest buffering capacity followed by cow milk and sheep milk based on the time it takes to reach a certain pH level [19]. Furthermore, in the case of dairy milk, individual constituents have varying levels of influence on buffering capacity. It has been reported that casein protein has an influence of 36%, whereas whey protein contributes 5.4%, despite both substances being proteins [20]. Similarly, it is believed that the proteins used in our experiments exhibited different relationships between pH and TA during fermentation as the proteins each exhibited unique buffering capacities. Furthermore, it has been reported that the buffering ability of a substance can undergo qualitative and quantitative changes in terms of buffer capacity or maximum buffering region upon heat treatment [21] as well as qualitative and quantitative changes following homogenization [22].

#### 3.2.2. Coagulum Strength and Viscosity

The coagulum strength that developed due to fermentation was also measured. Samples were prepared by storing the fermented broth produced after the end of the fermentation process in a cooler for three days. After cooling, a 15 mm press method was conducted. ISP produced a coagulum strength of 13.0 g_f_ and PP produced a higher coagulum strength of 47.5 g_f_ (Table 2). Furthermore, the coagulums that formed after fermentation were sieved with a 40 mesh and stored in a cooler to prepare samples for viscosity measurement. As a result, coagulums produced by protein coagulation due to fermentation were only observed in the ISP and PP samples. The highest viscosity resulting from coagulum fermentation was measured in the PP of 1932 cp (8 cp for pre-fermentation). This was followed by ISP with 1337 cp (10 cp for pre-fermentation), AP with 87 cp (17 cp for pre-fermentation), and RP with 16 cp (25 cp for pre-fermentation) (Table 2).

As the viscosities of ISP and PP were found to be greatly affected by pH (Figure 3), viscosity changes were measured at lower pH levels of 4.2–4.3 by extending the fermentation by an additional four hours (total 12 h) for ISP and PP. Both proteins exhibited significant viscosity changes due to fermentation (Figure 4). In the case of ISP, the extended fermentation resulted in a TA of 0.43% and a pH value of 4.27. The viscosity decreased compared to the viscosity after eight hours of fermentation to 463 cp (Figure 4). Additionally, syneresis also increased for ISP as the pH decreased, and the viscosity sharply decreased as the separated coagulums and liquid after cooling were homogenized and sieved.

With the extended fermentation, the TA and pH of the PP sample were determined as 0.37% and 4.26, respectively. Additionally, the viscosity of PP improved to 3013 cp compared to its eight-hour fermentation result (Figure 4). According to Klost and Drusch (2019) [16], PP gel formation occurs in two stages: as fermentation proceeds, a primary gel is formed at pH 6.0, and when the pH decreases to below 5.5, a secondary gel is formed as the charges of subunits approach the isoelectric point. However, an increase in viscosity was observed at around pH 3.5 due to the formation of the secondary gel in this case (Figure 3).

#### 3.2.3. Syneresis

As lactic acid is produced by LAB fermentation, the decreased pH alters the charge of the proteins, causing the proteins to coagulate and form gel-like coagulums. If coagulums formed by such a process have a low structural stability, this gives rise to the phenomenon known as syneresis, which is when a liquid (whey) is separated or extracted from a gel as the gel contracts [23]. From a consumer standpoint, syneresis and the difference in pH between the coagulums, as well as the discrepancies in whey measurement, reduce the quality of products. In this regard, syneresis is one of the most important quality indicators to consider in fermented milk production.

In the case of ISP, syneresis did not occur during storage as the homogenization improved the dispersibility. However, 5 mL of syneresis occurred as coagulums formed during fermentation (Figure 5). It has been reported that an increase in ionic strength, which results from the decreased pH during ISP fermentation, causes 11 S globulin to become unstable, resulting in coagulation [24,25]. Similarly, syneresis transpires due to the decreased water holding capacity of ISP during fermentation.

In the case of PP, syneresis did not occur both before and after fermentation (Figure 5). This may be related to the coagulum formation characteristics of PP according to pH: as the pH gradually decreased from the initial value of 7.3 due to the lactic acid produced during fermentation, coagulums began forming at around pH 6.2~6.0, greatly increasing the viscosity of the mixture. As a result, the increased viscosity allowed water to be effectively retained, preventing syneresis. It has been reported that biochemical processes, such as those involving LAB fermentation, yeast, or enzymes, can impart more favorable emulsification, viscosity, or creaming properties when manufacturing products using PP, producing stable products that are not vulnerable to syneresis [26].

Both RP and AP exhibited significant levels of syneresis regardless of whether fermentation was performed or not, and RP was found to be the most prone to syneresis (Figure 5). It has been reported that RP possesses the lowest solubility and the least favorable emulsification properties compared to other plant proteins [27]. In the case of AP, coagulums did not form during fermentation despite the decrease in pH, and there was a slight decrease in syneresis due to the fermentation process. We confirmed a small decrease from 17.5 mL before fermentation to 15.0 mL after fermentation.

### 3.3. Color

The color of a food product is a key design factor that significantly impacts a consumer’s choice through visual appeal even before tasting the product. Even in the case of fermented soy milk, which is the most common type of plant-based fermented milk being produced and researched, color is often cited as a major difference from conventional dairy milk. Compared to dairy milk, soy milk is darker in color. This has a negative effect in attracting consumers of fermented milk products made with regular dairy milk as it may evoke the unpleasant odors of soybeans [28]. Therefore, when testing various substances to improve the properties of fermented milk products, substances that produce a darker color than conventional soy milk should be avoided.

To compare the color differences of the samples, soy milk was set as a standard sample with the values of L = 84.8, a = 0.23, and b = 16.68. By using a color difference equation, ISP, which shares its source with soy milk, produced a color difference of 7.00, showing that it was most similar to the standard sample. In terms of similarity, ISP was followed by PP, which produced a color difference measurement of 13.48. The color differences of RP and AP were 25.06 and 31.59, respectively; these samples had a darker color and were the most dissimilar to the standard sample in terms of color (Table 3).

### 3.4. Physical Properties of Vegetable Dietary Fibers

#### 3.4.1. Syneresis and Viscosity

There are two main reasons behind the use of dietary fibers in fermented milk: to enforce properties by increasing the solid content and to improve properties through water binding. As AF is a water-soluble dietary fiber, syneresis could not be measured, as it is completely dissolved in the water, and physical changes such as increased viscosity were not observed. In the case of SF, syneresis occurred during storage due to the poor dispersibility of the insoluble constituents of SF before homogenization and pasteurization. After homogenization and pasteurization, the dispersibility of the constituents slightly improved, although 10 mL of syneresis occurred as precipitation took place during storage. Similar to SF, CF also exhibited syneresis in the stored sample before homogenization and pasteurization, but the viscosity rose to 2873 cp after homogenization and pasteurization, resulting in sticky, paste-like properties (Figure 6, Table 4). In addition to the altered properties and increased viscosity, syneresis did not occur during storage, indicating that CF is a suitable ingredient that can be used to achieve more favorable properties.

Numerous studies have reported findings regarding the increased viscosity and dispersion stability of CF through physical treatment. According to a report by Su et al. (2019) [29], subjecting CF to high-pressure homogenization treatment enhances its water-binding and water-holding capacities and also increases its viscosity. Another study [27] reported that homogenization changes the form of CF particles from granules to flakes, which results in increased viscosity due to the increased friction between CF particles. Furthermore, it was reported that viscosity is further increased if CF is subjected to greater pressures during the homogenization treatment [30].

#### 3.4.2. Soluble Content

Among the dietary fibers used in the experiment, AF is the most soluble, highlighted by the fact that it completely dissolved to produce a transparent solution and also produced the highest soluble content (4.70 Brix). Conversely, SF did not dissolve as it is an insoluble dietary fiber, resulting in an opaque appearance and the lowest soluble content (0.53 Brix). CF did not completely dissolve and resulted in a translucent appearance with a soluble content of 2.35Brix that lies between AF and SF. As such, CF was confirmed as the most suitable dietary fiber for fermented milk production as it does not result in syneresis due to property changes following homogenization and pasteurization. Furthermore, precipitation does not occur during storage.

Brix is a unit that indicates the amount of dissolved solute in an aqueous solution and is used to measure the “amount of soluble solids”. Examples of soluble solids include not only sugars, but also salts, proteins, and acids [31]. Therefore, a high Brix value does not simply reflect a high solid content. Among the various ingredients used to produce fermented milk, ingredients with low amounts of soluble solids form precipitates after 5–8 h, which is the general length of time of fermentation processes. This results in the irregular dispersion of the ingredient in the final product, giving rise to quality discrepancies between products manufactured in the early stages of production compared to those manufactured in the later stages. In severe cases, this can lead to quality issues such as accelerated whey off from the fermented coagulum. A wide array of research has reported solutions for issues arising from the implementation of ingredients with low solubility, such as the application of ultra-high-pressure homogenization (UHPH) to increase the dispersibility of the mixture and to increase structural stability [9,32,33]. However, conventional yogurt production processes are set with general high-pressure homogenization (HPH) facilities, which involve significantly lower pressures of 200 bar. Adding to the challenges is the high cost of installing plant-scale UHPH facilities for mass production, which renders it difficult to replace existing plant equipment for production. Therefore, the solubility of the ingredients is the most important quality factor to consider when selecting ingredients for the development of commercial products.

#### 3.4.3. Color

Soy milk was again used as the standard sample for the color difference test as in the case of the previous protein experiment. Using the color difference equation, the color difference of each dietary fiber sample was calculated (Table 5). AF showed the greatest color difference with a value of 76.55, followed by CF with 56.45 and SF with 19.61. SF was the most similar to the standard sample as it is also sourced from soy milk. As previously mentioned, the main reasons for adding dietary fiber to fermented milk are to reinforce properties by increasing solid content and to enhance properties through water binding. Although CF produced a darker color than the standard sample, CF is considered as the most suitable dietary fiber as it improves properties through physical treatment and does not result in syneresis.

## 4. Conclusions

This study investigated the properties of plant proteins and dietary fibers to select ingredients that can improve properties involved in the production of clean label stirred soy yogurt free of chemical additives. It was confirmed that the change in protein concentration did not affect the buffering capacity. After physical treatment, ISP, PP, and RP increased the buffer capacity with increasing the dB/dpH value, but in the case of AP, the buffer capacity decreased with the decreasing dB/dpH value. In the case of fermentation properties of the proteins, PP was evaluated as the most suitable protein to improve the properties of fermented milk as it developed a high coagulum strength upon fermentation. Furthermore, yogurts produced using PP had high viscosity and did not result in syneresis. In the case of dietary fibers, CF was deemed as the most suitable for fermented milk production as it increased the viscosity and avoided syneresis during storage after physical treatment. Consequently, this study concluded that PP and CF were the most suitable ingredients among the tested ingredients for the purpose of enhancing properties of fermented milk. By finding appropriate ratios of the two ingredients for use as property-enhancing agents, it may be possible to develop mixtures that are tailor-made for plant-based stirred soy yogurts.

## Figures and Tables

**Figure 1 foods-10-02305-f001:**
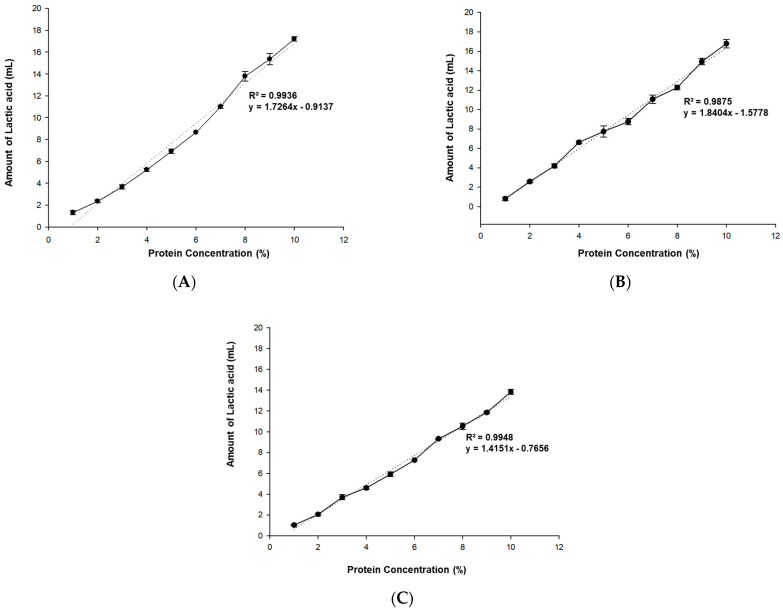
Titration capabilities of plant proteins: (**A**) isolated soy protein (ISP), (**B**) pea protein (PP), and (**C**) almond protein (AP).

**Figure 2 foods-10-02305-f002:**
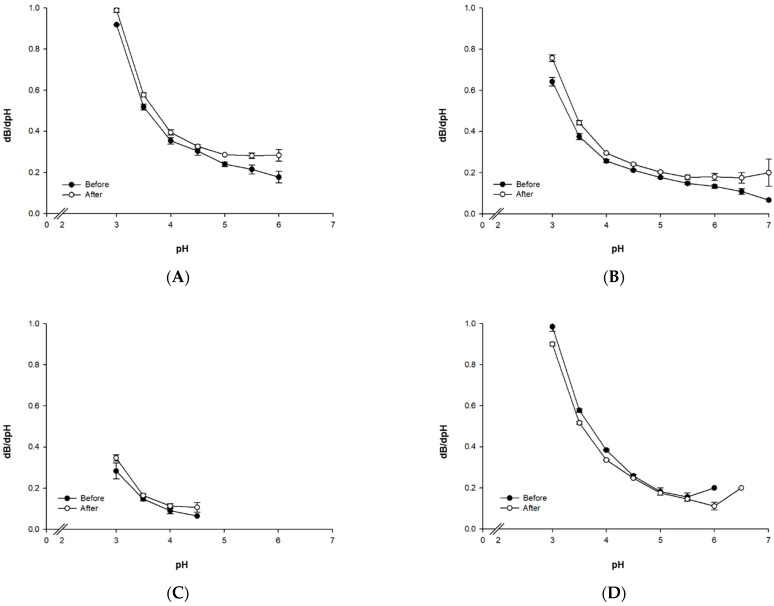
Buffering capacities of plant protein solutions (5%, *w*/*w*) before and after the physical treatment (pasteurization and homogenization): (**A**) isolated soy protein (ISP), (**B**) pea protein (PP), (**C**) rice protein (RP), and (**D**) almond protein (AP).

**Figure 3 foods-10-02305-f003:**
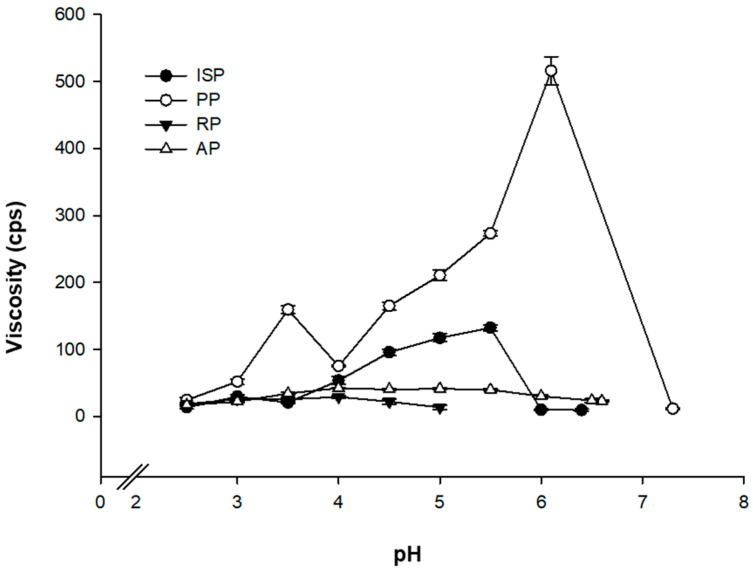
Apparent viscosities of plant protein solutions (5%, *w*/*w*) at different pH levels after the physical treatment (pasteurization and homogenization). ISP: isolated soy protein; PP: pea protein; RP: rice protein; AP: almond protein.

**Figure 4 foods-10-02305-f004:**
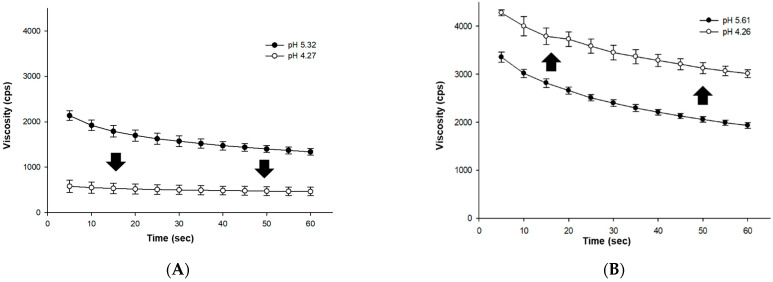
Effect of pH on the apparent viscosities of (**A**) ISP (isolated soy protein) and (**B**) PP (pea protein) solutions (5%, *w*/*w*) after 12 h of fermentation.

**Figure 5 foods-10-02305-f005:**
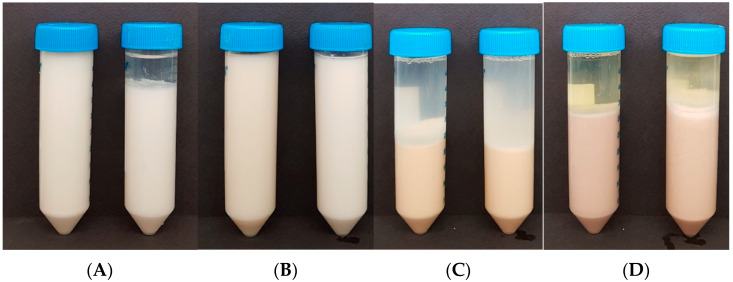
Syneresis of plant protein powder solutions (5%, *w*/*w*) before and after fermentation. (**A**) isolated soy protein (ISP), (**B**) pea protein (PP), (**C**) rice protein (RP), and (**D**) almond protein (AP). (left: before fermentation, right: after fermentation).

**Figure 6 foods-10-02305-f006:**
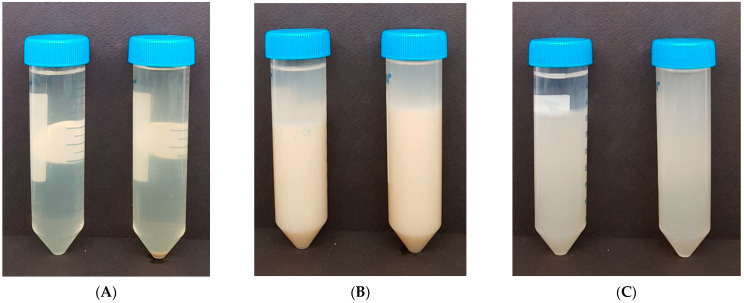
Syneresis of plant fiber solutions before and after the physical treatment (homogenization and pasteurization): (**A**) AF (acacia fiber, (**B**) SF (soy fiber), (**C**) CF (citrus fiber) (left: before physical treatment, right: after physical treatment).

**Table 1 foods-10-02305-t001:** Initial pH and 1 N lactic acid titration amounts to pH 3.0 of various vegetable protein solutions (5%, *w*/*w*) before and after the physical treatment (homogenization and pasteurization).

Protein	Before Physical Treatment		After Physical Treatment		Lactic Acid Ratio (% Increase)
	Initial pH	Lactic Acid (mL)	Initial pH	Lactic Acid (mL)	
ISP **	6.4	33.7 ± 0.8 ^b,^*	6.4	36.8 ± 0.6 ^a^	9.2
PP	7.3	32.4 ± 1.0 ^b^	7.3	38.0 ± 0.6 ^a^	17.3
RP	4.8	4.0 ± 0.5 ^d^	5.0	5.5 ± 0.2 ^d^	37.5
AP	6.1	30.4 ± 0.6 ^c^	6.6	33.8 ± 0.3 ^b^	11.2

* Means with different superscripts in the column significantly differ (*p* < 0.05). ** ISP: isolated soy protein; PP: pea protein; RP: rice protein; AP: almond protein.

**Table 2 foods-10-02305-t002:** Fermentation characteristics of various plant proteins (5%, *w*/*w*).

		ISP **	PP	RP	AP
pH		5.29 ± 0.03 ^b,^*	5.61 ± 0.04 ^a^	4.95 ± 0.03 ^c^	4.30 ± 0.04 ^d^
Titratable Acidity (%)		0.27 ± 0.02 ^a^	0.28 ± 0.03 ^a^	0.09 ± 0.03 ^b^	0.29 ± 0.04 ^a^
LAB *** (10 ^8^ cfu/g)		0.91 ± 0.12 ^b^	1.68 ± 0.11 ^a^	-	1.86 ± 0.20 ^a^
Coagulum strength (g_f_)		13.0 ± 1.91 ^b^	47.5 ± 2.41 ^a^	-	-
Viscosity (cp)		1337 ± 71 ^b^	1932 ± 59 ^a^	16 ± 2 ^c^	87 ± 1 ^c^
Syneresis (%)	Before fermentation	0.0	0.0	45.2 ± 0.4 ^a^	35.0 ± 0.6 ^b^
	After fermentation	10.0 ± 0.6 ^d^	0.0	45.0 ± 0.2 ^a^	30.0 ± 0.8 ^c^

* Means with different superscripts in the row significantly differ (*p* < 0.05). ** ISP: isolated soy protein; PP: pea protein; RP: rice protein; AP: almond protein. *** LAB: Lactic Acid Bacteria.

**Table 3 foods-10-02305-t003:** Color of plant protein solutions (5%, *w*/*w*) after fermentation.

Properties	Soy Milk	ISP **	PP	RP	AP
L	84.80	78.24 ± 0.01 ^a,^*	71.45 ± 0.01 ^b^	64.31 ± 0.02 ^c^	55.17 ± 0.02 ^d^
a	0.23	0.07 ± 0.01 ^d^	1.77 ± 0.01 ^c^	8.15 ± 0.02 ^b^	9.65 ± 0.01 ^a^
b	16.68	14.25 ± 0.03 ^d^	17.52 ± 0.02 ^c^	28.74 ± 0.03 ^a^	22.23 ± 0.02 ^b^
ΔE ***		7.00 ± 0.00 ^d^	13.48 ± 0.01 ^c^	25.06 ± 0.01 ^b^	31.59 ± 0.01 ^a^
WI ***	77.43	73.99 ± 0.02 ^a^	66.44 ± 0.01 ^b^	53.46 ± 0.00 ^c^	49.04 ± 0.01 ^d^

* Means with different superscripts in the row significantly differ (*p* < 0.05). ** ISP: isolated soy protein; PP: pea protein; RP: rice protein; AP: almond protein *** ΔE and WI: absolute color difference calculated from L, a, b.

**Table 4 foods-10-02305-t004:** Physical properties of vegetable dietary fibers.

Properties	SF **	AF	CF
Syneresis (%)	20.0 ± 1.2	- ***	0
Viscosity (cp)	85 ± 4 ^b,^*	14 ± 6 ^c^	2873 ± 8 ^a^
Soluble content (Brix)	0.52 ± 0.01 ^c^	4.70 ± 0.07 ^a^	2.35 ± 0.03 ^b^

* Means with different superscripts in the row significantly differ (*p* < 0.05). ** SF: soy fiber; AF: acacia fiber; CF: citrus fiber, *** Syneresis could not be measured for AF as it is a water-soluble dietary fiber.

**Table 5 foods-10-02305-t005:** Color of plant dietary fibers (5%, *w*/*w*).

Properties	Soy Milk	AF **	SF	CF
L	84.80	10.95 ± 0.04 ^c,^*	65.25 ± 0.04 ^a^	31.26 ± 0.03 ^b^
a	0.23	−1.18 ± 0.01 ^c^	0.08 ± 0.02 ^a^	−1.07 ± 0.03 ^b^
b	16.68	−3.40 ± 0.04 ^c^	15.15 ± 0.03 ^a^	−1.15 ± 0.02 ^b^
ΔE ***		76.55 ± 0.03 ^a^	19.61 ± 0.03 ^c^	56.45 ± 0.04 ^b^
WI ***	62.09	10.87 ± 0.04 ^c^	62.09 ± 0.04 ^a^	31.24 ± 0.03 ^b^

* Means with different superscripts in the row significantly differ (*p* < 0.05). ** AF: acacia fiber; SF: soy fiber; CF: citrus fiber. *** ΔE and WI: absolute color difference calculated from L, a, b.

## Data Availability

Not applicable.

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
