# Peer review of "Applicable Plant Proteins and Dietary Fibers for Simulate Plant-Based Yogurts"

_foods, 2021, doi:10.3390/foods10102305_

Round 1

Reviewer 1 Report

The manuscript presents many interesting results. The introduction focuses unnecessarily only on Korean market data, whereas word data would be preferred. Moreover, the introduction lacks many references (indicated in detailed comments). Materials and methods are unclear, it seems that authors have only tested protein or fiber solutions/suspensions/colloidal systems. On the other hand the article focuses on vegetable/soy milk yoghurt. Maybe an experiment flowchart would solve this problem. Some of the experiments seem ununnecessarily overbuild i.e. titration capability of proteins – it is obvious that it will increase linearly with concentration. The discussion in most parts is conducted properly. The conclusion are supported by the results but not related directly to the title of the manuscript. Literature should adopted to the MDPI requirements. I suggest revision of the manuscript by the authors with emphasize to clearly present the flow of the experiment and its results.

Some of selected issued that should be addressed:

Line 15-16 viscosity in titration?

Line 17 why solubility is a problem?

Line 26-34 no source for the whole paragraph, moreover why Korean market is discussed only?

Line 35-36 those trends are common in most developed countries

Line 43-44 source should be provided

Line 47 treatment with what?

Line 50 response surface methodology (abbreviation was not explained earlier)

Line 66 water content? water biding would be more suitable

Line 57-70 no source was provided for the whole paragraph

Line 88 strain or culture?

Line 107 fermented products do not require sterilization, pasteurization is sufficient

Line 116 95°C is not sterilization

Line 193 why syneresis was measured without centrifugation?

Line 404 probably not solely by pH decrease did the viscosity change

Line 447 colour value? Or rather absolute color difference

Line 458 table with synereis in % rather than photograph should be placed

Line 461-469 source should be provided

Author Response

Please find an attached response to reviewer.

Reviewer 2 Report

Comments and recommendations:

Please, modify (rephrase) sentences: l. 47-54, 107-108, 121-122, 150-152, 360-363, 428-431, 568-570

Please unify 1N or 1 N

In the chapter Material and Methods, I did not find the methods of LAB determination (cfu/g)

Tables and Figures:

  • Please explain all abbreviations in the footer of the table (Table 2 – Table 5, Figure 3, Figure 5)
  • Please, use Italic letter for p
  • Please, indicate better the statistical significance:

Means with different superscripts in the column significantly differ (p < 0.05) (Table 1)

Means with different superscripts in the row significantly differ (p < 0.05) (Table 2-5)

References – should be improved (typographical errors etc.)

Other comments:

  • 45-46 – … the effects of soy yogurt; only a few studies …
  • 50 – explain abbreviation “RSM”
  • 102 – A series of suspensions were prepared …
  • 110-111 – A portion of the prepared suspensions was placed … and was used …
  • 237-238 – This is also true …
  • 240 – … a strong correlation is likely between the protein powder …
  • 245, 246 – livestock species
  • 247 – Similarly, plant phosphate content …
  • 257 – … to the physical treatment compared to ISP and PP.
  • 308 – … amandine ?
  • 432 – … of the most important quality indicators to consider …
  • 498 – Another study [27] reported that homogenization …
  • 516 – CF was confirmed as the most suitable …
  • 537 – … is the most important quality factor …
  • 545 – … the main reasons of (or for?) adding …

Author Response

(The authors gave the same response as above.)

Reviewer 3 Report

The manuscpript is focused on the possible use of plant proteins and fibers to be used as ingredients for plant-based yogurts. the methodology and the different analysis performed are positive points but the infromation derived are not always useful or the best chosen to achieve the aim of the work. The terminology involving dairy products is often used as inappropriate as it is very different compared to plant proteins and derived products. The study is however interesting but far from an application, it is a laboratory characterization which is still far from any application or even to mention finished products, so i will adjust title and discussion accordingly

The title has to be changed in my opinion because it is a preliminary study on vegetable proteins to simulate plant-based yogurts.

- line 42-43: the sentence has to be inverted: The demand for plant-based yogurts is rising as an alternative to yogurts made with milk.

- line 110: the concentration of 5% w/w refers to powder or protein concentration? since protein concentration is different among the powders it should be specified

- line 121: replace “that is produced due to” with “by”

- line 150: please delete “on the lactic acid titration volume”

- line 182: curd strength is inappropriate as curd derives essentially from rennet coagulation of milk. it is better defined in coagulum. Correct here and elsewhere in the manuscript.

- line 198-200: this sentence seems a repetition and still it is not clear how syneresis was determined. please correct

- lines 223-225: this is not clear to me, please explain better or avoid this assumption

- lines 371-374. This is not acceptable. As you know, the formation of milk coagulum can be obtained from the action of rennet (enzymatic coagulation) or from fermentation (acid coagulation) which is this case. Coagulation is achieved during fermentation because the lactic acid produced lowers the pH until it reaches the isoelectric point of the milk proteins, resulting in aggregation. Therefore fermentation is not directly producing a coagulum in any substrate, such as the plant-based proteins you are investigating.  

line 392: curd strength in this study can be deleted as it is meaningless. the measurement of the apparent viscosity is enough to describe the system

- line 575: “derived from soy milk” should be deleted. It is better to talk about soy yogurt or soy beverage rather than soy milk. Milk is specifically defined from mammalian milk

Table 1. Lactic acid ratio could be more effectively represented by % increase of lactic acid before and after

Author Response

(The authors gave the same response as above.)

Round 2

Reviewer 1 Report

The manuscript has been substantially improved. I would like to thank the authors for thorough review. However, I have few minor remarks that should be addressed.

Line 18 solubility of a hydrocolloid is not a problem it still binds water, please rephrase

Line 491 (Table 3) and Line 566 (Table 5) I think the manuscript would benefit from calculation of whiteness index (the formula can be found in https://www.mdpi.com/1420-3049/26/17/5417/htm )

Line 518 – value of syneresis would be better represented if recalculated to %

Line 520 water soluble hydrocolloids can still be characterized by syneresis, the problem with AF it that it is hardly visible (but separation can be spotted with bare eye on the photograph provided), this was also the reason why I suggested centrifugation of the samples.

Author Response

Response to Reviewer1 2nd round

The manuscript has been substantially improved. I would like to thank the authors for thorough review. However, I have few minor remarks that should be addressed.

Line 18 solubility of a hydrocolloid is not a problem it still binds water, please rephrase

Ans) Thank you for your suggestion. We rephrased them as you suggested. Please see line 17-19 in the revised manuscript.

Line 491 (Table 3) and Line 566 (Table 5) I think the manuscript would benefit from calculation of whiteness index (the formula can be found in https://www.mdpi.com/1420-3049/26/17/5417/htm )

Ans) Thank you for your suggestion. We added whitness index as you suggested. Please see line 210-216, line 500 (Table 3) and line 575 (Table 5) in the revised manuscript.

Line 518 – value of syneresis would be better represented if recalculated to %

Ans) Thank you for your suggestion. We added syneresis (%) as you suggested. Please see line 369 (Table 2) and line 500 (Table 3) in the revised manuscript.

Line 520 water soluble hydrocolloids can still be characterized by syneresis, the problem with AF it that it is hardly visible (but separation can be spotted with bare eye on the photograph provided), this was also the reason why I suggested centrifugation of the samples.

Ans) Thank you for your suggestion. We mentioned that AF was soluble and cannot characterize syneresis in line 508-510 in the revised manuscript.

Reviewer 3 Report

The authors replied to almost all the critical points and the manuscript was improved. I suggest to delete or sum up in one sentence the conclusions from line 573 to 582.

Author Response

Response to Reviewer 3 round 2

The authors replied to almost all the critical points and the manuscript was improved. I suggest to delete or sum up in one sentence the conclusions from line 573 to 582.

Ans) Thank you for your suggestion. We corrected sonclusion as you suggested. Please see line 581-584 in the revised manuscript.